# Exploring pathogenesis and biomarkers through establishment of a rat model of male infertility with liver depression and kidney deficiency

Ying Shen [1,2,3,4], Jian Fan[1], Shaobo Liu[1,2,3], Ling Tao[1,2,3]*, Qingbo Yang[4]*, Xiangchun Shen[1,2,3]*

1 The State Key Laboratory of Functions and Applications of Medicinal Plants, Guizhou Medical University, Guian New District, Guiyang, Guizhou, China, 2 The High Efficacy Application of Natural Medicinal Resources Engineering Center of Guizhou Province, School of Pharmaceutical Sciences, Guizhou Medical University, Guian New District, Guiyang, Guizhou, China, 3 The Key Laboratory of Optimal Utilization of Natural Medicine Resources, School of Pharmaceutical Sciences, Guizhou Medical University, Guian New District, Guiyang, Guizhou, China, 4 The National Engineering Research Center of Miao's Medicines, Guizhou Yibai Pharmaceutical Co., Ltd., Yunyan District, Guiyang, Guizhou, China

* 649511230@qq.com (LT); 67510872@qq.com (QY); shenxiangchun@126.com (XS)

**Data Availability Statement:** All relevant data are within the manuscript and its Supporting Information files.

## Abstract

### Objectives

To establish a rat model that accurately replicates the clinical characteristics of male infertility (MI) with Liver Depression and Kidney Deficiency (LD & KD) and investigate the pathogenesis.

### Methods

After subjecting the rats to chronic restraint stress (CRS) and adenine treatment, a series of tests were conducted, including ethological assessments, evaluations of reproductive characteristics, measurements of biochemical parameters, histopathological examinations, and analyses of urinary metabolites. Additionally, bioinformatics predictions were performed for comprehensive analysis.

### Results

Compared to the control, the model exhibited significant manifestations of MI with LD & KD, including reduced responsiveness, diminished frequency of capturing estrous female rats, and absence of mounting behavior. Additionally, the kidney coefficient increased markedly, while the coefficients of the testis and epididymis decreased significantly. Sperm counts and viabilities decreased notably, accompanied by an increase in sperm abnormalities. Dysregulation of reproductive hormone levels in the serum was observed, accompanied by an upregulation of proinflammatory cytokines expressions in the liver and kidney, as well as exacerbated oxidative stress in the penile corpus cavernosum and testis. The seminiferous tubules in the testis exhibited a loose arrangement, loss of germ cells, and infiltration of

**Funding:** This work was supported by the Scientific Research Foundation for Advanced Talents, Guizhou Medical University [Grant No. 26232020130].

**Competing interests:** The authors have declared that no competing interests exist.

inflammatory cells. Furthermore, utilizing urinary metabolomics and bioinformatics analysis, 5 key biomarkers and 2 crucial targets most closely linked to MI were revealed.

## Conclusion

The study successfully established a clinically relevant animal model of MI with LD & KD. It elucidates the pathogenesis of the condition, identifies key biomarkers and targets, and provides a robust scientific foundation for the prediction, diagnosis, and treatment of MI with LD & KD.

## 1. Introduction

In recent years, the rise in unhealthy lifestyles and habits, such as excessive smoking, heavy alcohol consumption, chronic sleep deprivation, frequent sauna usage, and heightened psychological stress, has led to a surge in infertility, which has emerged as the third most intricate health condition globally after cardiovascular diseases and tumors, posing a significant threat to human reproductive health [1]. According to pertinent statistics, 15% to 18% of couples within the childbearing age may encounter infertility challenges, with male factors contributing to nearly 50% of cases [2, 3]. Alarmingly, the burden of infertility affects at least 30 million men worldwide [4]. Male infertility (MI) is a multifactorial disorder resulting in male reproductive dysfunction, primarily manifesting when couples of childbearing age, engaging in regular sexual activity without contraception for over a year, find the female partner unable to conceive due to male-related factors [5]. Unfortunately, the lives of patients have been significantly compromised due to the impact of personal privacy concerns and the protracted nature of the treatment procedures [6].

The pathogenesis of male infertility (MI) has yet to be definitively elucidated, and its etiological factors are also multifaceted. Within modern medicine, prevailing viewpoints indicate that the principal causative factors encompass testicular dysfunction, aberrant spermatogenic cells, compromised sperm motility, disturbances in reproductive endocrine regulation, and other related elements [7]. In the realm of clinical practice, various diagnostic tests are employed to determine the presence of MI and ascertain its underlying causes. For example, sperm quality can be evaluated by assessing motility and morphology rates [8]. Analyzing physiological markers such as prolactin (PRL), gonadotropin releasing hormone (GnRH), follicle stimulating hormone (FSH), and testosterone (T) can offer insights into gonadal function [9]. In specific circumstances, a biopsy of reproductive tissues like the testes can be performed to assess cellular health and histopathological changes [10]. By employing these diagnostic methodologies, it is feasible to ascertain, to a certain extent, the presence of infertility in males. Regrettably, these techniques can also carry certain negative implications. While the collection of semen samples for analysis through masturbation is a common clinical approach, the context of medical masturbation might induce emotional distress or psychological resistance, leading to diminished libido in patients. Moreover, a testicular biopsy can result in transient discomfort, localized bleeding and swelling, and even the potential for infection, all of which can impact fertility.

Over thousands of years of inheritance and development, traditional Chinese medicine (TCM) has garnered significant advantages in the understanding of MI and offers unique insights into its etiology and pathogenesis [11]. Within TCM, the five visceral organs, including the liver, heart, spleen, lungs, and kidneys, represent not just anatomical structures but

encompass individual physiological and psychological systems that collectively constitute the entire body [12]. Based on TCM, the liver plays a pivotal role in blood storage, stress relief, and emotional regulation, with Liver Depression being linked to impotence. Modern medical research also acknowledges that depression can lead to sexual dysfunction [13]. Long-term depression can impact the normal metabolism of the body, leading to a certain degree of endocrine imbalance. It affects the secretion of androgens, hindering the normal movement of sperm and resulting in infertility [14]. Similarly, the kidney is believed to store essence and control reproduction, whereas Kidney Deficiency might result in compromised sperm production or reduced sperm quality [15]. In TCM theory, it is stated that the liver and kidneys share a common origin, and essence and blood mutually support each other. The liver is responsible for storing blood and nourishing the external kidneys. If there is stagnation of liver Qi due to liver depression, and it loses its function of smoothly regulating the distribution of Qi and blood, then Qi and blood cannot effectively nourish the external kidneys and the renal system, thus potentially inducing kidney deficiency. Consequently, it is observed that MI is often associated with Liver Depression and Kidney Deficiency (LD & KD), making them common clinical syndromes in MI [16]. In clinical practice, key symptoms of Liver Depression include anxiety, depression, and reduced appetite, whereas Kidney Deficiency is marked by increased urine output, sensitivity to colds, and lower back soreness. To establish a diagnosis, it is imperative to not only identify these symptoms but also integrate findings from contemporary medical evaluations, enabling a comprehensive analysis of etiology and pathogenesis, thereby facilitating the formulation of personalized treatment strategies for MI.

Establishing an animal model of MI that accurately reflects the syndrome characteristics of LD & KD not only facilitates a deep understanding of the pathogenesis of infertility but also holds immeasurable significance for disease prediction and diagnosis. Furthermore, the model supports the integration of Chinese and Western medicine practices, promotes the modern validation of TCM theory, and offers a more comprehensive exploration of the relationship between LD & KD and reproductive system dysfunction, which provides novel insights for MI treatment strategies.

In establishing an animal model of MI characterized by LD & KD, a comprehensive evaluation is essential. For instance, general behavioral observations, including fur condition, mental state, food and water intake, urine output, body weight, and spontaneous activity in rodents, are important criteria for assessing the successful establishment of the model. Additionally, sperm quality, erectile function, and sexual behavior are widely recognized standards for evaluating MI models. Furthermore, various physiological and biochemical indices such as reproductive hormones, inflammatory factors, and oxidative stress cannot be overlooked as reference standards. To simulate Liver Depression syndrome in TCM, several animal modeling methods are available, including chronic restraint stress (CRS), hormonal intervention, and surgical trauma, among others [17]. CRS, among these methods, refers to the simulation of prolonged and continuous physiological stress stimuli. This approach authentically replicates common stressors encountered in human daily life, such as work-related pressures and environmental fluctuations. As a result, it more accurately emulates the compromised hepatic Qi and blood circulation status, comprehensively reflecting the intricate pathological mechanisms of liver depression syndrome [18]. Moreover, the simplicity and feasibility of CRS operations reduce dependency on complex equipment and techniques, and it provides non-invasive stimulation to the animal subjects' physiology. Additionally, its strong reproducibility enables consistent replication across different research laboratories, enhancing the reliability and comparability of research findings [19]. Administering specific drugs, adenosine, to influence kidney function and induce experimental animals into a state of Kidney Deficiency is currently a primary method for generating infertility [20, 21]. On one hand, adenosine possesses

reproductive function impairing effects, simulating the impact of purine metabolism disruptions on kidney function, effectively replicating the physiological characteristics of Kidney Deficiency. On the other hand, the effects induced by adenosine not only remain confined to the kidneys but trigger multisystem physiological alterations throughout the entire organism, aligning with the multisystem impairment features of Kidney Deficiency. It is regrettable that a singular modeling approach falls short of adequately reflecting the clinical manifestations of MI. Exploring and establishing a comprehensive animal model that encompasses multifactorial causes of infertility, more closely mirroring clinical reality, like LD & KD, holds important research value and clinical significance.

Metabolomics, which focuses on endogenous small-molecule metabolites within the organism and investigates the comprehensive dynamic patterns of their types and quantity changes, aligns with the holistic perspective of TCM theory and has been widely applied in the TCM field for the discovery of disease biomarkers, evaluation of pharmacological effects and toxicity, and exploration of mechanisms of action [22, 23]. In consideration of this, the present study utilizes a combination of factors (CRS and adenosine) to establish a rat model of infertility, conducting a comprehensive evaluation encompassing ethological analysis, biochemical indices, organ coefficients, and histopathology to assess the established model's alignment with clinical features of MI with LD & KD in both TCM and modern medicine. Leveraging ultra-performance liquid chromatography-quadrupole-time-of-flight mass spectrometry (UPLC-Q-TOF-MS), differences in small-molecule metabolites within urine samples of normal rats and rats with MI of LD & KD are compared, aiming to identify biomarkers associated with LD & KD and unravel pivotal metabolic pathways, thus elucidating the biological mechanisms of MI. The objective is to provide a scientific foundation for the establishment and application of an animal model of MI with LD & KD, as well as for research on the treatment of the disease.

## 2. Materials and methods

### 2.1 Animals and experimental design

A total of 24 male Sprague Dawley (SD) rats of SPF grade were provided by the Experimental Animal Center of Guizhou Medical University, with a body weight of (200 ± 20) g. Subsequently, the rats were subjected to a randomized allocation procedure, leading to the formation of two distinct experimental groups, namely, the control and the model, each comprising 12 rats. Additionally, 12 female SD rats within the body weight range of 180 g to 220 g were also supplied by the Experimental Animal Center of Guizhou Medical University. The rats were housed in a well-ventilated environment, with the room temperature maintained at 23~25°C, humidity ranging from 50% to 70%, and a 12-hour light-dark cycle. All animal experimentation procedures received approval from the Ethics Committee of Experimental Animals at Guizhou Medical University (Approval No. 2200367) and were carried out rigorously in adherence to national regulations on experimental animal ethics.

After 7 days of acclimatization, male rats in the model were induced to develop MI with LD & KD using CRS in combination with adenine. The specific protocol involved placing the rats in a restraint device with an adjustable length of 20 cm, an outer diameter of 7 cm, and an inner diameter of 5 cm, thereby limiting their free movement for 8 hours daily [24]. Following the completion of restraint, the rats were subjected to intragastric administration of an adenine solution (prepared using sodium carboxymethyl cellulose) at a dose of 300 mg·kg$^{-1}$·d$^{-1}$, with a volume of 0.5 mL·(100 g)$^{-1}$, continuously for 30 days. In contrast, the control rats were allowed unrestricted movement and received an equivalent volume of sodium carboxymethyl cellulose through intragastric administration.

## 2.2 Ethological monitoring

**2.2.1 Spontaneous activity and physical signs.** Throughout the experiment, it is necessary to daily observe factors including the condition of the rats' fur, mental state, and spontaneous activity. Moreover, measurements should be recorded for parameters including food intake, water consumption, urinary output, as well as body weight.

**2.2.2 Sucrose preference test.** On the 27th day of modeling, 6 rats were randomly selected from both the control and the model, respectively. The rats underwent a 3-day adaptation period involving the provision of sucrose solution, distilled water, and an appropriate amount of food. During this period, CRS and adenine were administered continuously. On the 30th day, following a 24-hour fasting and water restriction period, the sucrose preference test was conducted as follows [25]: Each rat was provided with a bottle containing 100 mL of sucrose solution and another bottle containing 100 mL of distilled water. Subsequently, the amount of sucrose solution consumed by the rat within a 1-hour timeframe was recorded to calculate the sucrose preference index, and the index was calculated using the formula: Preference Index = [Amount of Sucrose Solution Consumed / (Amount of Distilled Water Consumed + Amount of Sucrose Solution Consumed)] × 100%.

**2.2.3 Sexual ethological assessment.** After completing the sucrose preference test, male rats were utilized for the male-female mating experiment. To prepare female rats, subcutaneous injections of estradiol benzoate (100 $\mu g \cdot kg^{-1}$) and progesterone (2 $mg \cdot kg^{-1}$) were administered at 48 hours and 4 hours before the mating experiment, respectively, to induce estrus. Both the male and the female were co-housed within one cage, where their interactions were observed and recorded for 30 minutes in a dimly lit and observable quiet environment, with particular attention to mounting attempts, intromission events, and latency periods [26]. Considering the potential impact of sexual ethological experiments on male hormonal levels and physiological metabolism, the male post-sexual ethological was not employed for further investigations.

## 2.3 Biochemical indices and organ coefficients measurements

**2.3.1 Reproductive hormones.** After completing 30 days of modeling, anesthesia was induced through intraperitoneal injection of sodium pentobarbital (30 $mg \cdot kg^{-1}$), followed by euthanasia using the same agent upon termination of the experiment. Blood was collected from the abdominal aorta and centrifuged at 3500 $r \cdot min^{-1}$ for 10 minutes to obtain serum, which was then stored at -80°C. Serum levels of PRL, GnRH, FSH, luteinizing hormone (LH), T, and estradiol ($E_2$) were measured using the enzyme-linked immunosorbent assay (ELISA).

**2.3.2 Neurotransmitter.** After obtaining serum, it was stored at -80°C. The levels of 5-hydroxytryptamine (5-HT) and norepinephrine (NE) in the serum were measured using ELISA.

**2.3.3 Organ coefficients.** Following blood collection, the kidneys, testes, and epididymides were harvested to determine the organ coefficients. The coefficients were calculated using the formula: Organ Coefficient = (Organ Mass / Body Mass) × 100%.

**2.3.4 Proinflammatory cytokines.** The left lateral lobe of the liver and the left kidney were initially rinsed with physiological saline and blotted dry with filter paper. Subsequently, they were fixed in 4% paraformaldehyde. Immunohistochemical analysis was employed to measure the levels of interleukin-6 (IL-6), tumor necrosis factor-alpha (TNF-α), and interleukin-1 beta (IL-1β) within the liver (kidney).

**2.3.5 Sperm parameters.** The left epididymis was placed in a disposable culture dish containing 5 mL of pre-warmed (37°C) PBS buffer. The epididymis was finely minced using tissue scissors, gently agitated, and then incubated at 37°C for 2 minutes to facilitate the release of

sperm. Sperm count was determined using a hemocytometer. Following staining with eosin, the viability and morphological abnormalities of sperm were observed.

**2.3.6 Neurological enzymes.** After the removal of the cartilage from the penis, blood residues were carefully removed, followed by fixation with 4% paraformaldehyde. Immunohistochemical analysis was employed to assess the presence of neuronal nitric oxide synthase (nNOS) and endothelial nitric oxide synthase (eNOS) within the penile corpus cavernosum.

**2.3.7 Oxidative stress markers.** Under ice bath conditions, the left testis was homogenized with 5% physiological saline, followed by centrifugation at 3000 r·min$^{-1}$ for 10 minutes, and the supernatant was collected and stored at -80˚C. Specific assay kits were employed to determine the levels of superoxide dismutase (SOD), glutathione peroxidase (GSH-Px), and malondialdehyde (MDA) within the testis.

## 2.4 Histopathological Examination

The right testis was fixed in 4% paraformaldehyde and underwent standard procedures of dehydration, clarification, paraffin embedding, sectioning, mounting, staining, and drying. Hematoxylin and eosin staining were used for the histopathological assessment of testicular morphological alterations under light microscopy.

## 2.5 Urine metabolomics analysis

**2.5.1 Sample collection and processing.** Following the completion of modeling, urine was collected over 12 hours. Subsequently, the urine was subjected to centrifugation at 13,000 r·min$^{-1}$ for 15 minutes under refrigerated conditions (4˚C), and the supernatant was aspirated and diluted fourfold with distilled water, followed by a 30-second vortex mixing step to ensure homogeneity. Afterward, the diluted samples were filtered through a 0.22 μm microporous membrane and the filtrate was utilized for UPLC-Q-TOF-MS analysis.

**2.5.2 Detection conditions.** Separation using the ACQUITY™ UPLC BEH Amide column, the mobile phase composed of a 25 mmol·L$^{-1}$ ammonium acetate aqueous solution and acetonitrile (A) was employed for gradient elution, and the sequence consisted of 95% A for 0 to 0.5 minutes, followed by a transition from 95% A to 65% A during 0.5 to 7 minutes, further decreasing to 40% A from 7 to 8 minutes, maintaining 40% A from 8 to 9 minutes, and then transitioning from 40% A to 95% A between 9 to 9.1 minutes, which was followed by a continuous elution at 95% A from 9.1 to 12 minutes. A flow rate of 0.5 mL·min$^{-1}$ was maintained throughout the analysis. The column was thermostatically controlled at 25˚C, while the sample compartment was maintained at 10˚C. An injection volume of 2 μL was used for sample introduction.

Following separation by the Agilent 1290 Infinity LC UPLC, samples were subjected to mass spectrometric analysis using the Triple TOF 6600. Detection was carried out in both positive and negative ion modes using electrospray ionization, and 10 fragment spectra were collected for each scan.

**2.5.3 Statistical analysis.** Using the aforementioned analytical conditions, whole-scan data acquisition was performed on the urine samples from both the control and the model. In the next step, the acquired MS data were then imported into Progenesis QI 2.0 for data preprocessing, encompassing tasks like peak matching, extraction, and normalization. Following this, unsupervised Principal Component Analysis (PCA) was conducted applying EZinfo 2.0 to distinguish differences between the control and the model. Furthermore, Orthogonal Partial Least Squares Discriminant Analysis (OPLS-DA) was employed to assess the reliability and stability of the animal model, and Variable Importance in Projection was computed. Concurrently, inter-group ion normalized abundances were calculated, and t-tests were applied to filter out ions exhibiting significant differences ($P < 0.05$).

By integrating MS/MS information with resources such as the Human Metabolome Database, Lipid Maps, and Kyoto Encyclopedia of Genes and Genomes (KEGG), potential biomarkers were identified. Subsequently, the identified biomarkers underwent metabolic pathway enrichment analysis using Metabo Analyst 5.0. This comprehensive process ultimately resulted in the creation of an intuitive and interactive exploratory data network system.

**2.5.4 Selection of disease core targets and molecular docking with key biomarkers.** Utilizing urinary metabolomics analysis-derived pivotal biomarkers as key search terms, relevant targets of these biomarkers were gathered from the TCM Systems Pharmacology Database and Analysis Platform and Swiss TargetPrediction. Employing "male infertility" as the focal keyword, disease-associated targets were compiled from the GeneCards, Online Mendelian Inheritance in Man, and Disease Gene Network. Subsequently, a protein-protein interaction (PPI) network was constructed through the Search Tool for the Retrieval of Interacting Genes/Proteins, and the visual representation and network topology analysis of the PPI network was accomplished using Cytoscape 3.8.0, thus facilitating the identification and selection of core targets.

The SDF files of key biomarkers and the PDB files of core targets were initially retrieved separately from the Public Chemical Database and Protein Data Bank. Subsequently, utilizing Autodock Vina 1.2.2, a preprocessing phase was executed to prepare the obtained files, which encompassed the execution of a comprehensive all-atom docking process. Following the docking procedure, the resultant data were visualized using PyMOL, facilitating the generation of a graphical representation of the docking outcomes. Ultimately, pivotal biomarkers exhibiting the highest relevance to MI were identified through this systematic process.

## 2.6 Molecular dynamics simulation (MDS) [27, 28]

Gromacs 2022.3 software was used for MDS. For small molecule preprocessing, AmberTools22 is used to add GAFF force field to small molecules, while Gaussian 16W is used to hydrogenate small molecules and calculate RESP potential. Potential data will be added to the topology file of the molecular dynamics system. The simulation conditions were carried out at a static temperature of 300K and atmospheric pressure (1 Bar). Amber99sb-ildn was used as a force field, water molecules were used as a solvent (Tip3p water model), and the total charge of the simulation system was neutralized by adding an appropriate number of Na+ ions. The simulation system adopts the steepest descent method to minimize the energy, and then carries out the isothermal isovolumic ensemble equilibrium and isothermal isobaric ensemble equilibrium for 100000 steps, respectively, with the coupling constant of 0.1 ps and the duration of 100ps. Finally, the free MDS was performed. The process consisted of 5000000 steps, the step length was 2fs, and the total duration was 100ns. After the simulation was completed, the built-in tool of the software was used to analyze the trajectory, and the root-mean-square variance (RMSD), root-mean-square fluctuation (RMSF), and protein rotation radius of each amino acid trajectory were calculated, combined with the free energy molecular mechanics poisson-boltzmann surface area (MM/PBSA) and other data.

## 3. Results

### 3.1 Alterations in ethology, organic coefficients, sperm parameters, and testicular pathology

From the 5th day of modeling, rats in the model exhibited increased water consumption and urine output. By the 12th day, a reduction in the rate of body mass gain and decreased spontaneous activity were observed. Furthermore, these conditions worsened with the prolongation

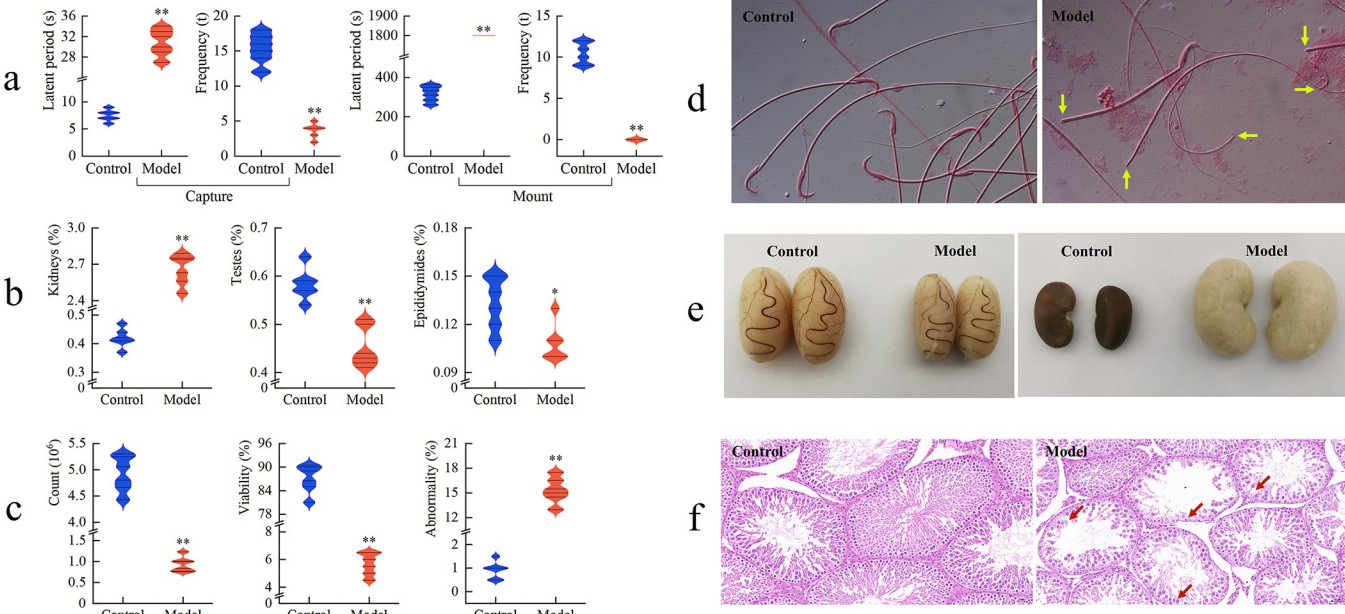

**Fig 1. Reproductive characteristics (n = 6).** a: sexual behavior of male rats towards estrous female rats, male rats subjected to CRS and adenine treatment exhibited reduced spontaneous activity, decreased frequencies in capturing and mounting estrous females, and a significantly prolonged latency period; b: organic coefficients of kidneys, testes, and epididymides, depicted significantly increased kidney coefficients, indicating aversion to cold, increased urine volume, and reduced libido associated with adenine, additionally, epididymis coefficients decreased, suggesting damage to reproductive organs; c: sperm parameters; d: sperm morphology after eosin staining (100×), after CRS and adenine treatment, there was a significant decrease in both sperm count and viability, accompanied by a notable increase in sperm abnormality, these results emphasize the detrimental impact of CRS and adenine on male reproductive function; e: the morphology of the kidneys and testis; f: histopathologyl of testes stained with hematoxylin and eosin (20×), following CRS and adenine treatment, the testicular volume of rats significantly shrank, seminiferous tubules narrowed with reduced germ cells and disordered arrangement, along with infiltration of inflammatory cells. This indicates testicular damage, further confirming the adverse effects of CRS and adenine on male reproductive function.

of the modeling period, and on the 25th day, symptoms characteristic of LD & KD started to manifest, including delayed responsiveness, squinting, piloerection, huddling, and dry and sparse fur. In contrast, rats in the control did not display significant changes in food intake, water consumption, activity, responsiveness, or fur quality. In terms of sucrose preference, the model exhibited a lower level, at approximately 70.12%, whereas the control displayed a higher level, at about 96.12%. Regarding sexual behavior, as illustrated in Fig 1A, when contrasted with the control, the male rats in the model demonstrated an exceedingly significant prolongation of the latent period for capturing estrous female rats (P < 0.01), coupled with a markedly substantial decrease in capture frequency (P < 0.01). Additionally, no mounting behavior was observed within the 1800-second timeframe.

As evidenced by the data presented in Fig 1B, the kidneys coefficient in the model exhibited a markedly significant increase (P < 0.01), approximately 6.27 times greater than that of the control. Conversely, the testes coefficient demonstrated a highly significant decrease (P < 0.01), while the epididymides coefficient, despite displaying a decreasing trend, did not reach statistical significance (P > 0.05).

With respect to sperm parameters, as evidenced by Fig 1C, in comparison to the control, the model exhibited a highly significant decrease in both count and viability (P < 0.01), along with a markedly significant increase in abnormality (P < 0.01). Furthermore, a substantial number of sperm exhibited anomalies such as headless, tailless, or coiled tails, as indicated by the yellow arrows in Fig 1D.

Compared with the control group, the kidney volume of the model significantly increased and turned white, commonly known as "big white kidney", and the testicular volume

significantly shrunk, with narrow and pale seminiferous tubules, as shown in Fig 1E. Upon examination under 20x magnification, the tunica albuginea of the control displayed structural integrity, a substantial number of seminiferous tubules with clear and regular boundaries, a compact arrangement of germ cells including elongated spermatozoa, and there was no proliferation of interstitial cells within the testicular stroma, and no infiltration of inflammatory cells was observed. In contrast, the model demonstrated a more loosely arranged pattern of seminiferous tubules, characterized by conspicuous atrophy of the germinal epithelium, reduced counts of germ cells at various developmental stages, along a minor amount of germ cell shedding (indicated by red arrows), and localized interstitial edema, as depicted in Fig 1F.

## 3.2 Variations in biochemical indices within serum, penile corpus cavernosum, testis, liver, and kidney

Fig 2A presents the levels of reproductive hormones in serum, demonstrating a significant difference between the control and the model ($P < 0.01$). In the model, LH, FSH, and PRL showed significant increases, while GnRH, T, and $E_2$ exhibited significant decreases. These alterations indicate that the model has experienced hormonal imbalance.

After the establishment of the model, the levels of 5-HT and NE in the serum are depicted in Fig 2B. When compared to the control, the serum levels of 5-HT and NE in the model showed a significant reduction ($P < 0.01$), with the decrease in 5-HT being greater than that of NE. From the perspective of neurotransmitter downregulation, this indicates the successful establishment of Liver Depression.

As illustrated in Fig 2C, following the modeling procedure, the positive expression of eNOS and nNOS in the penile corpus cavernosum significantly decreased (indicated by the tan or yellow staining), particularly in the case of nNOS, as compared to the control. This may suggest a compromised erectile function in the penises of the model rats.

In comparison to the control, the activities of SOD and GSH-Px in the model significantly decreased, while the MDA content showed a marked increase, as evidenced by the data presented in Fig 2D. This reveals that the testes of the model rats have undergone oxidative stress damage, leading to a concomitant decrease in antioxidative capacity.

Based on Fig 2E, it is evident that, in comparison to the control, the model exhibited a significant increase in the positive expression of proinflammatory cytokines, such as IL-6, IL-1β, and TNF-α, within the liver (depicted in shades of tan or yellow). Moreover, the density of positive cells expressing proinflammatory cytokines in the kidney also showed a noteworthy increase, although to a lesser extent than observed in the liver. Collectively, these findings indicate that the liver and kidney of the rats, particularly the liver, have exhibited significant inflammatory responses as a consequence of the prolonged influence of CRS in conjunction with adenine.

## 3.3 Changes in urinary metabolites

**3.3.1 Metabolic profile of endogenous metabolites.** By superimposing the total ion chromatograms of each sample for comparative spectral overlay, the results, as depicted in Fig 3A, demonstrate a substantial congruence in the response intensities and retention times of individual chromatographic peaks. This congruence underscores the minimal variability attributable to instrumental error throughout the entirety of the experimental procedure.

All identified endogenous metabolites (in both positive and negative ion modes) were categorized based on the chemical taxonomy, as demonstrated in Fig 3B. Among these, organic acids and derivatives constituted the predominant proportion, accounting for around 38.24%, followed by nucleosides, nucleotides, and analogs, which accounted for approximately 15.44%.

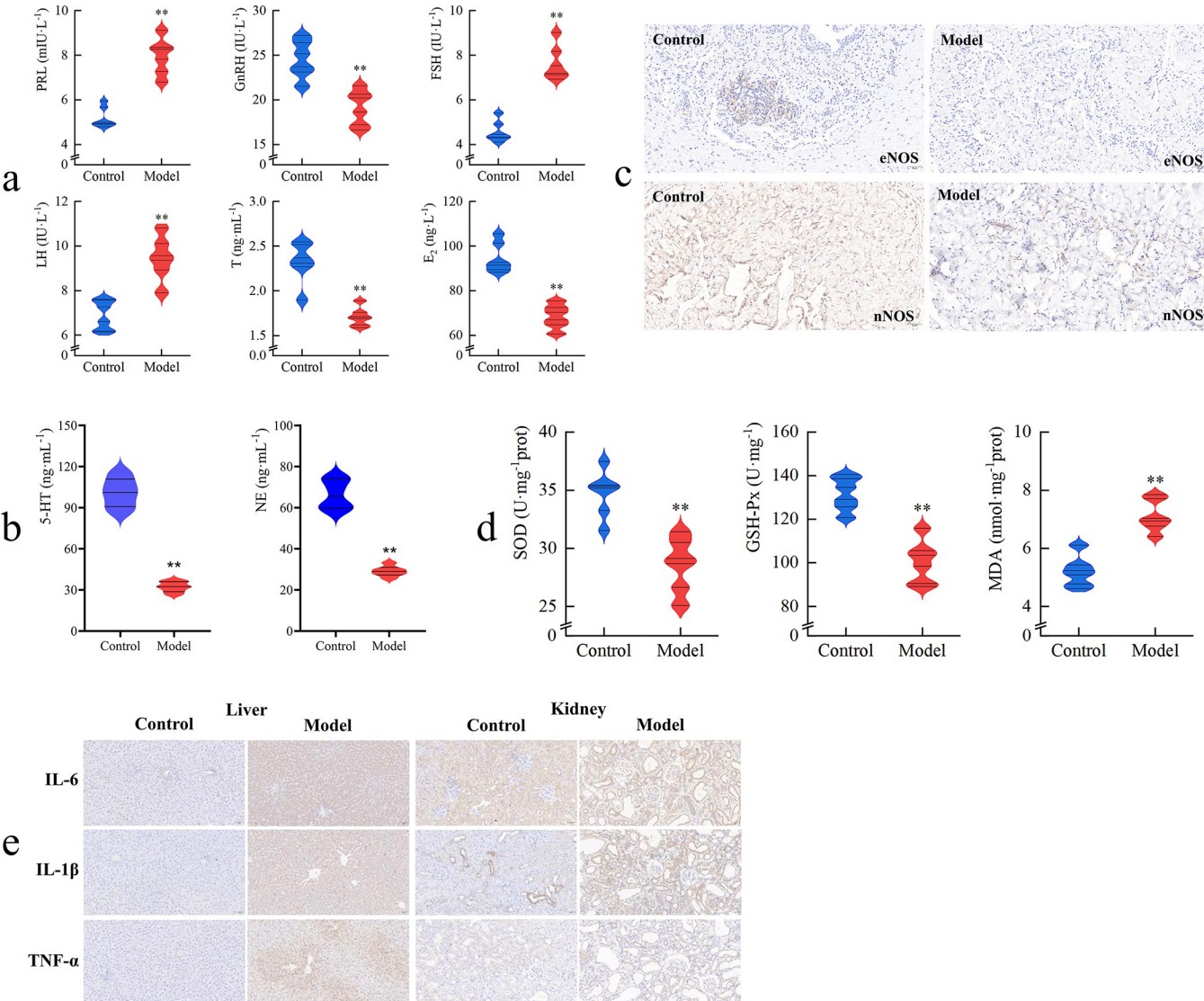

**Fig 2. Determination of biochemical indicators (n = 6).** a: levels of reproductive hormones in serum, in the model, LH, FSH, and PRL levels significantly increased, while GnRH, T, and E$_2$ levels decreased, indicating a hormonal imbalance compared to controls; b: changes in neurotransmitters in serum, serum levels of 5-HT and NE significantly decreased in the model, with 5-HT reduction being greater than NE. This suggests successful establishment of Liver Depression, characterized by neurotransmitter downregulation; c: positive expressions of eNOS and nNOS in the penile corpus cavernosum (20×), positive expression of eNOS and nNOS in the penile corpus cavernosum significantly decreased in the model particularly nNOS, indicating compromised erectile function; d: activities of SOD and GSH-Px and MDA content in the testes, SOD and GSH-Px activities significantly decreased, while MDA content markedly increased in the model, indicating oxidative stress damage to the testes and reduced antioxidative capacity; e: positive expression of proinflammatory cytokines in the Liver And Kidney (20×), the model exhibited a significant increase in proinflammatory cytokine expression, such as IL-6, IL-1β, and TNF-α, in the liver and kidneys compared to controls. This suggests significant inflammatory responses in these organs, particularly in the liver, due to prolonged CRS and adenine exposure.

Through data analysis, metabolic profiling information of rat urine was acquired, as depicted in Fig 3C. It is evident that within both positive and negative ion modes, there is distinct intra-group clustering of endogenous metabolites between the model and the control, with prominent inter-group differentiation. Simultaneously, as illustrated in Fig 3D, the OPLS-DA model effectively discriminates between samples from the control and the model.

**3.3.2 Potential biomarkers.** By contrasting the differential endogenous metabolites in the urine of the control and the model, a total of 136 potential biomarkers were identified for MI

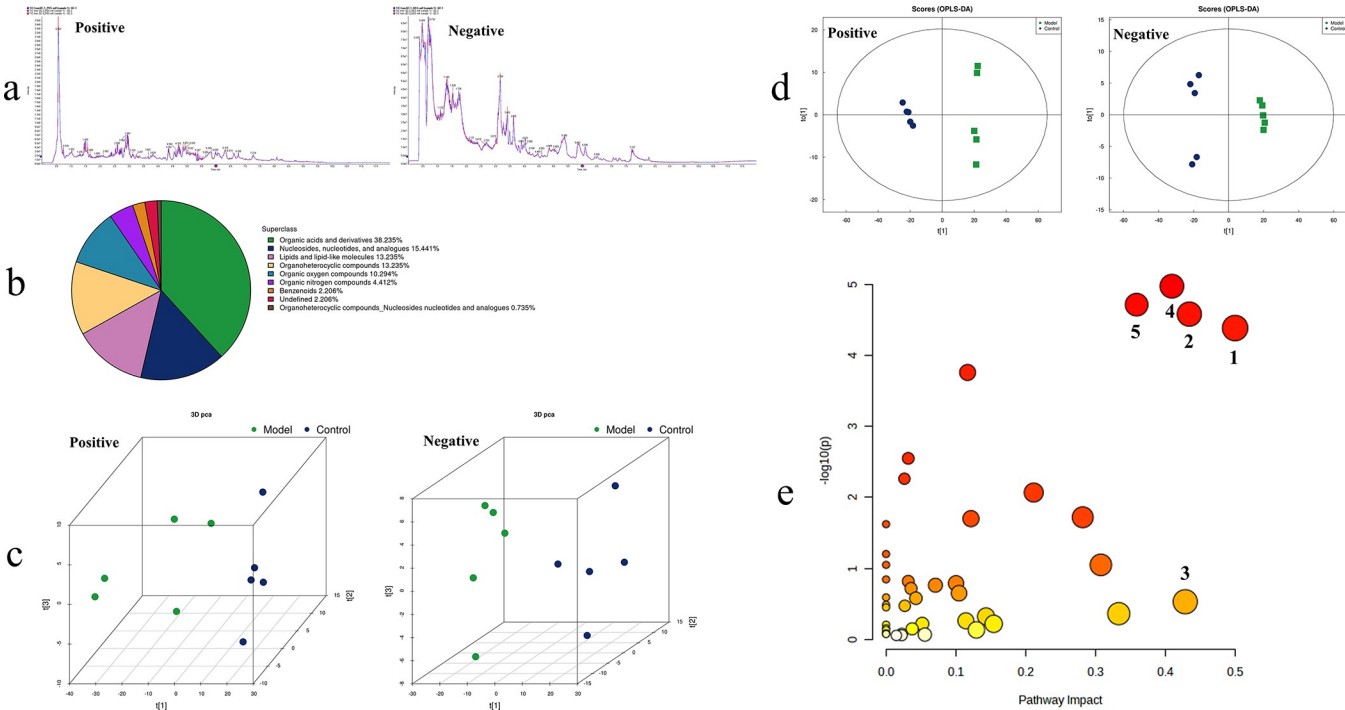

**Fig 3. Information on endogenous metabolites in urine (n = 6).** a: total ion current; b: chemical taxonomy, endogenous metabolites were categorized based on chemical taxonomy, with organic acids and derivatives constituting approximately 38.24%, followed by nucleosides, nucleotides, and analogs at around 15.44%; c: PCA, metabolic profiling of rat urine shows distinct clustering within both positive and negative ion modes, with clear differentiation between the model and control groups; d: OPLS-DA, differentiating the control from the model based on metabolic data; e: KEGG pathway enrichment analysis of urinary biomarkers. Positive: positive ion mode; Negative: negative ion mode. 1: D-glutamine and D-glutamate metabolism; 2: histidine metabolism; 3: taurine and hypotaurine metabolism; 4: alanine, aspartate and glutamate metabolism; 5: pyrimidine metabolism.

with LD & KD, as detailed in S1 File. Among these, 69 were identified under the positive ion, and 67 were identified under the negative ion.

**3.3.3 Key biomarkers and metabolic pathways.** Retrieving the information of the aforementioned 136 potential biomarkers, and subsequently conducting pathway enrichment analysis, yielded pertinent metabolic pathways, as presented in Fig 3E. Following this, employing a selection criterion of impact factor greater than 0.35, relevant metabolic pathways associated with MI of LD & KD were identified, comprising five main pathways (P < 0.01). These pathways include the D-glutamine and D-glutamate metabolism, histidine metabolism, taurine and hypotaurine metabolism, alanine, aspartate, and glutamate metabolism, as well as pyrimidine metabolism. A total of 23 pivotal biomarkers are documented, as detailed in Table 1. When compared to the control, 13 biomarkers in the model exhibited an upward trend, including succinate, glutamine, fumarate, histidine, thymine, succinic semialdehyde, 2'-deoxycytidine 5'-monophosphate (dCMP), deoxythymidine 5'-phosphate (dTMP), 2'-deoxyuridine 5'-monophosphate (dUMP), D-glutamine, orotidine, 4-imidazoleacetic acid, and 3-methyl-L-histidine, while the remaining 10 biomarkers exhibited a downward trend.

**3.3.4 Central targets intersected by key biomarkers and clinical MI.** Through relevant queries, a total of 190 associated functional targets for the 23 key biomarkers were identified. Simultaneously, utilizing "male infertility" as the keyword yielded 3189 relevant functional targets. Following intersection analysis, a final set of 51 shared targets between the two datasets was obtained (refer to Fig 4A). Detailed information is provided in S2 File.

As depicted in Fig 4B, the constructed PPI network encompasses 51 nodes and 201 edges, and the mean node degree within the cohort of 51 targets is 7.77. Among these, the top 5

**Table 1. Metabolic pathway information of urinary biomarkers.**

| No. | Metabolic Pathway | Metabolite Count | Metabolite Names | Impact Value | -lg$P$ |
|---|---|---|---|---|---|
| 1 | D-glutamine and D-glutamate | 4 | D-glutamine (↑), L-glutamine (↓), glutamic acid (↓), alpha-ketoglutarate (↓) | 0.50 | 4.38 |
| 2 | histidine | 6 | histidine (↑), 4-imidazoleacrylic acid (↓), 3-methyl-L-histidine (↑), glutamic acid (↓), 1-methylhistamine (↓), 4-imidazoleacetic acid (↑) | 0.43 | 4.58 |
| 3 | taurine and hypotaurine | 1 | taurine (↓) | 0.43 | 0.53 |
| 4 | alanine, aspartate and glutamate | 8 | citrate (↓), glutamic acid (↓), alpha-ketoglutarate (↓), glutamine (↑), L-asparagine (↓), fumarate (↑), succinic semialdehyde (↑), succinate (↑) | 0.41 | 4.97 |
| 5 | pyrimidine | 9 | orotidine (↑), glutamine (↑), cytidine (↓), dCMP (↑), dUMP (↑), deoxycytidine (↓), dTMP (↑), 3-ureidopropionic acid (↓), thymine (↑) | 0.36 | 4.71 |

↑: upregulated; ↓: downregulated.

ranked targets are glyceraldehyde-3-phosphate dehydrogenase (GAPDH), SRC proto-oncogene, non-receptor tyrosine kinase (SRC), epidermal growth factor receptor (EGFR), caspase 3 (CASP3), and prostaglandin-endoperoxide synthase 2 (PTGS2), with the degrees of 26, 20, 19, 19, and 18, respectively. Comprehensive details are available in S3 File.

**3.3.5 Molecular docking between key biomarkers and core targets.** The binding interactions between key biomarkers and core targets were investigated through the constructed molecular docking models. Using binding free energies ($\leq$ -6 kJ·mol$^{-1}$) and the number of interaction bonds ($\geq$ 2) as screening criteria, the 5 most pivotal biomarkers were identified. It

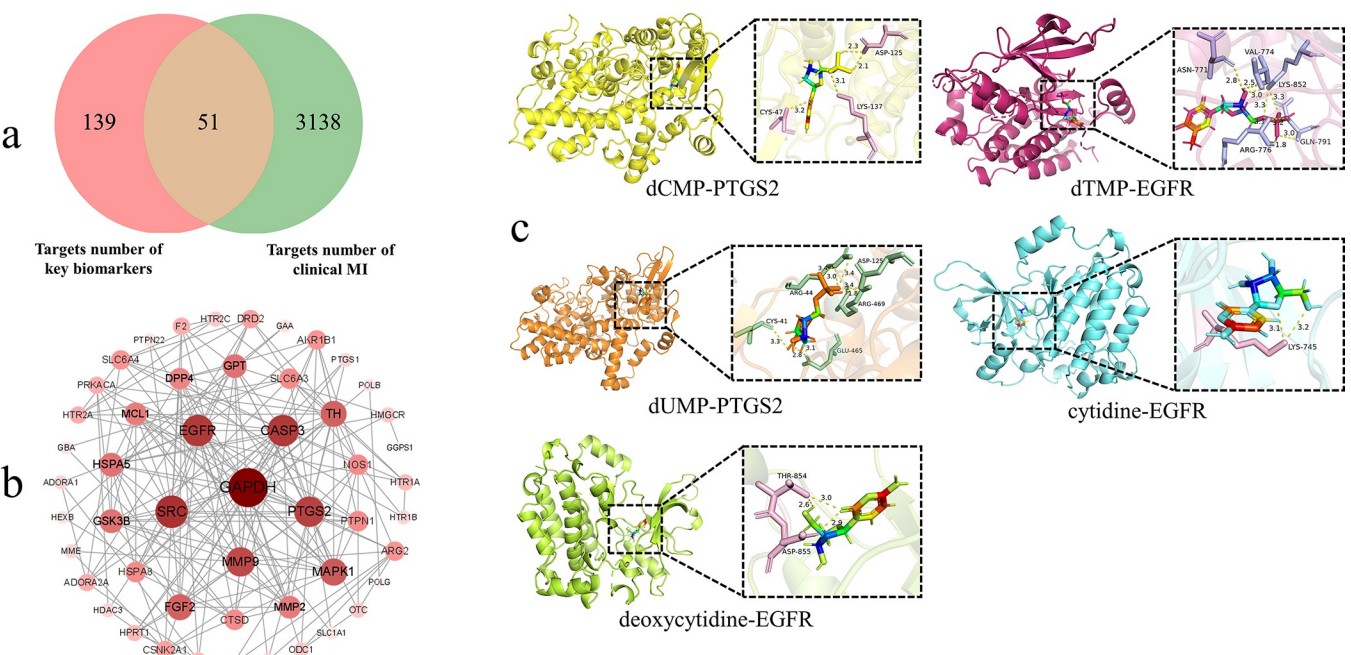

**Fig 4. Relevant information on key targets and biomarkers docking.** a: venn of intersecting targets number, a total of 190 targets associated with key biomarkers were identified, along with 3,189 targets linked to clinical MI, with an overlap of 51 targets; b: PPI network of shared targets. Nodes represent targets, edges represent interactions between targets, and node size and color correspond to the degree. Through the PPI network, the top 5 ranked targets were identified as GAPDH, SRC, EGFR, CASP3, and PTGS2. c: molecular docking between the most pivotal biomarkers and the central targets. Using molecular docking models, 5 pivotal biomarkers were identified based on binding free energies ($\leq$ -6 kJ·mol$^{-1}$) and interaction bond criteria ($\geq$ 2). Specific bond interactions were observed between these biomarkers and their optimal targets.

is noteworthy that the docking information between these 5 biomarkers and their optimal targets can be observed in Fig 4C. The pertinent information regarding the optimal docking outcomes is as follows: there are 4 bonds (-6.84 kcal·mol$^{-1}$) between dCMP and PTGS2's CYS 47, LYS 137, and ASP 125; dTMP and EGFR display 9 bonds (-6.27 kcal·mol$^{-1}$) involving ASN 771, VAL 774, LYS 852, ARG 776, and GLN 791; dUMP presents 8 bonds (-7.10 kcal·mol$^{-1}$) with PTGS2's CYS 41, GLU 465, ARG 469, ASP 125, and ARG 44; Cytidine demonstrates 2 bonds (-6.10 kcal·mol$^{-1}$) with EGFR's LYS 745; Deoxycytidine features 3 bonds (-6.57 kcal·mol$^{-1}$) with EGFR's THR 854 and ASP 885.

## 3.4 MDS analysis

Firstly, the RMSD of the protein-ligand complex was calculated during the simulation process to assess its structural stability (Fig 5A). The trend in the radius of gyration (Rg) of the protein was analyzed to evaluate the compactness of the protein structure during the simulation (Fig 5C). An analysis of the solvent-accessible surface area (SASA) provided a comprehensive assessment of the system, describing the extent of solvent exposure of the protein-ligand complex during the simulation (Fig 5D). From the RMSD, Rg, and SASA plots, it is evident that the system reached equilibrium and maintained stability at around 60 nanoseconds of simulation. This indicates a favorable stability of the binding between EGFR, PTGS2, and the metabolite small molecules, with a compact structure and relatively stable interaction with the solvent.

To further evaluate the fluctuation of the protein during the simulation, the RMSF values of each residue were computed (Fig 5B). The plots reveal significant fluctuations around residues 200 and 80 for EGFR and PTGS2, respectively, suggesting that these regions may be crucial for the protein's structure and function, indicating a higher degree of flexibility.

Additionally, hydrogen bonds (HBond) were assessed as an indicator of the number of hydrogen bonds within the protein. As shown in Fig 5E, a substantial number of hydrogen

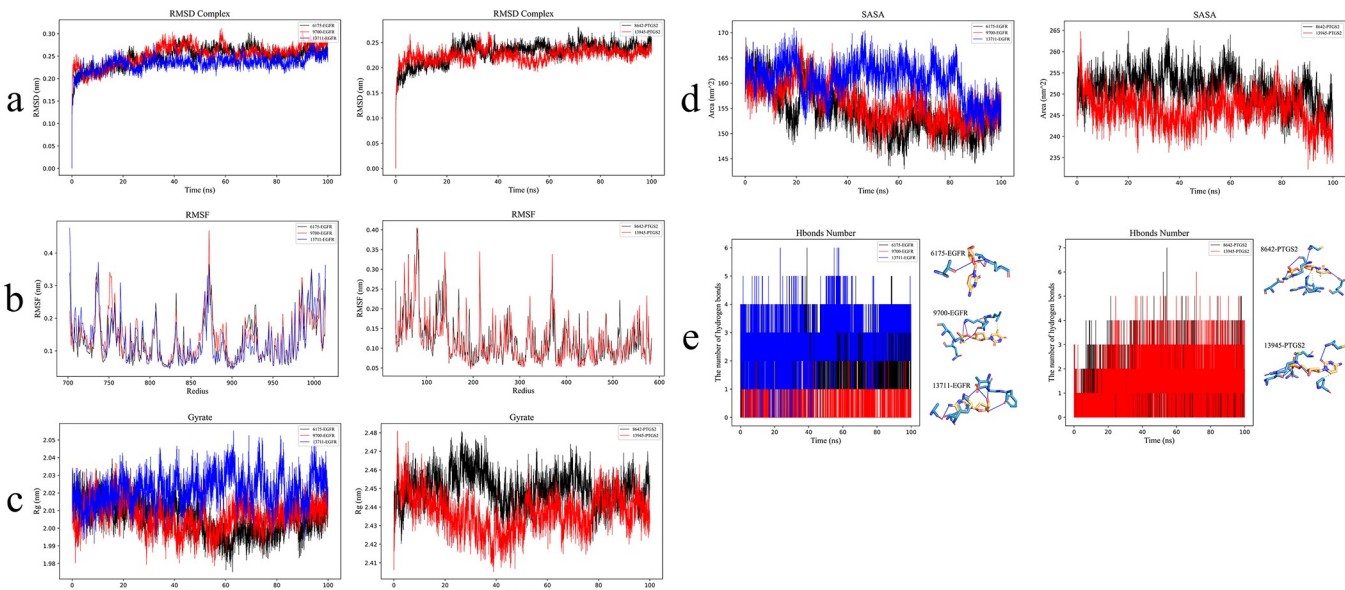

**Fig 5. Analysis of MDS results.** a: RMSD of the ligand complex; b: RMSF of the ligand complex; c: Total duration of Rg for the ligand complex; d: Solvent accessible surface area plot of the ligand complex; e: Number of hydrogen bonds between the protein and small molecules. 6175: Cytidine, 9700: dTMP, 13711: Deoxycytidine, 8642: dUMP, 13945: dCMP. The binding of EGFR, PTGS2, and five metabolite small molecules demonstrated good structural stability, fluctuation characteristics, and energy stability.

**Table 2. Binding free energies of the five complex systems.** (kcal/mol).

|  | Cytidine-EGFR | dTMP-EGFR | Deoxycytidine-EGFR | dUMP-PTGS2 | dCMP-PTGS2 |
|---|---|---|---|---|---|
| ΔVDWAALS | -34.72 | -17.22 | -37.75 | -37.23 | -39.60 |
| ΔEEL | -31.99 | 0.46 | -28.07 | -74.01 | 0.00 |
| ΔEGB | 42.69 | 18.81 | 34.82 | 61.15 | 13.53 |
| ΔESURF | -4.75 | -2.25 | -4.68 | -5.57 | -4.68 |
| ΔGGAS | -66.71 | -16.76 | -65.81 | -111.25 | -39.60 |
| ΔGSOLV | 37.93 | 16.56 | 30.14 | 55.58 | 8.85 |
| ΔTOTAL | -28.78 | -0.21 | -35.67 | -55.67 | -30.76 |

bonds were observed between the protein and small molecules during the simulation, indicating strong stability in their interactions.

To determine the reasonableness of the binding between the protein and small molecules, the widely used MM/PBSA analysis was conducted. By calculating the energy of the protein-small molecule complexes, the total free energy of the complex was obtained, allowing for an assessment of the stability of the protein-small molecule interactions. As shown in Table 2, the binding energy values within the range of -0.21 to -55.67 suggest a stable interaction between the protein and small molecules. In summary, through MDS and the analysis of related indicators, it was observed that the binding of EGFR, PTGS2, and five metabolite small molecules exhibited good structural stability, fluctuation characteristics, and energy equilibrium.

## 4. Discussion

Clinical investigations have revealed that LD & KD stands as the most prevalent syndrome among MI [29]. Leveraging cutting-edge methodologies of contemporary scientific inquiry, coupled with advanced research strategies, to establish an animal model exhibiting parallels or comparability with MI, while concurrently identifying pivotal biomarkers. This endeavor, whether directed at gaining profound insights into the etiology of MI or towards the realms of prognosis, diagnosis, and therapeutic interventions, holds immeasurable significance.

### 4.1 Construction model of MI with LD & KD based on CRS and adenine, and evaluation of ethological and reproductive traits

In the process of constructing animal models, the observation of ethological features serves as a foundational cornerstone for research. Consequently, the precise documentation and assessment of ethological changes in rats emerge as crucial during the model establishment phase [30]. Research findings unveil that male rats, following CRS and adenine treatments, exhibit a series of ethological traits associated with LD & KD. These traits include heightened urine excretion, reduced appetite, delayed responses, increased water consumption, aversion to cold, as well as a sparse and lackluster appearance of fur, among others. Furthermore, after repeated restraint stress, male rats demonstrate phenomena such as relinquished struggling and a gradual or even diminished weight gain.

To assess hedonic deficits in animals, a sucrose preference test was employed, serving as an effective ethological assessment and currently being one of the most reliable methodologies, capable of detecting the degree of depression [31]. Research findings demonstrate that male rats, after CRS and adenine treatments, exhibited a significant reduction in sucrose preference, indicative of a potential depressive state in these animals. Furthermore, the male rats in the model displayed decreased spontaneous activity, markedly reduced frequencies in capturing and mounting estrous females, and a considerably prolonged latency period (as depicted in

Fig 1A). This series of phenomena further elucidates that CRS and adenine could lead to a notable decline in the reproductive capacity of male rats.

With the continuous application of CRS, the activity of male rats gradually shifts from an active state to an inhibited one. They transition from resisting the restraint procedure to accepting it, exhibiting delayed responses, which align with the development of a depressive state. These outcomes indicate that CRS induces rats to manifest behaviors resembling depression. Subsequent research reveals a substantial increase in immobility time among rats after 21 days of CRS [32]. Following the modeling process, male rats exhibit reduced sucrose preference and diminished spontaneous activity, thereby revealing depressive symptoms in these animals attributable to CRS. Depression, known as Liver Depression in TCM, adversely impacts male reproductive functions, emerging as a notable contributor to MI [33]. Moreover, the integration of ethological traits and the markedly increased kidney coefficients (review the data outlined in Fig 1B) suggests that the observed phenomena in male rats of the model, such as aversion to cold, increased urine volume, and reduced libido, are associated with Kidney Deficiency, aligning with the action of adenine. Collectively, the animal model established through the utilization of CRS and adenine successfully simulates the clinically observed symptoms of LD & KD.

Sperm parameters, such as sperm count, viability, and abnormality, constitute the fundamental indicators for evaluating male reproductive function, serving as the most prevalent diagnostic tools to identify MI [34]. Sperm count reflects the testicular spermatogenic function, while viability and abnormality more directly reflect sperm quality. Research findings reveal a markedly significant decrease in both sperm count and viability, coupled with a notably substantial increase in sperm abnormality, observed in rats subjected to CRS and adenine treatment (details can be found in Fig 1C and 1D). This outcome further underscores the detrimental impact of CRS and adenine on the reproductive function of male rats [35].

The testes serve as the primary organs for spermatogenesis and the secretion of male hormones, while the epididymides act as the developmental and mature "cradle" for sperm, possessing absorption, secretion, and concentration functions [36]. In other words, the testes are responsible for sperm generation and hormone secretion, while the epididymides play a pivotal role in sperm maturation, storage, and provision of an appropriate environment. The coordination between the two ensures the normal functionality of the male reproductive system. Put differently, abnormalities in the testes and epididymides could lead to reduced sexual function and even MI [37]. Research findings demonstrate that after CRS and adenine treatment, the testicular volume of rats notably shrank, the seminiferous tubules became narrower, and germ cells decreased with disordered arrangement, accompanied by the infiltration of inflammatory cells, as depicted in Fig 1F. Furthermore, the epididymis coefficients also diminished (check Fig 1B). This series of phenomena indicates that the reproductive organs of the model male rats have incurred damage, further corroborating that the combination of CRS and adenine can exert adverse effects on male reproductive function.

In summary, the combination of CRS and adenine can induce damage to male reproductive function. Alongside the assessment of ethological and reproductive traits, the established animal model accurately mirrors the clinical features of LD & KD, providing a dependable platform for further investigation into MI with LD & KD.

## 4.2 Pathogenesis of MI with LD & KD

**4.2.1 Oxidative stress and endocrine imbalance as precursors to MI.** Penile erectile dysfunction is a prominent manifestation of male sexual inadequacy. As a pivotal regulatory pathway, nitric oxide (NO) plays a crucial role in maintaining the process of penile erection [38].

NO, an endogenous signaling molecule, is catalyzed by nitric oxide synthase (NOS) and facilitates the accelerated influx of blood into the penile arteries and cavernous sinusoids, thereby inducing penile erection [39, 40]. The classification of NOS encompasses constitutive types (cNOS) and inducible types (iNOS), with the former further divided into neuronal (nNOS) and endothelial (eNOS) isoforms, both of which contribute to the regulation of penile erection [41]. Research findings reveal that under the influence of CRS and adenine, the positive expression of nNOS and eNOS within the penile corpus cavernosum significantly decreases, notably nNOS (consult information in Fig 2C), leading to adverse effects on the erectile function of male rats. This is further corroborated by behavioral outcomes, such as reduced mounting frequency and prolonged latent period in response to estrous females. This dynamic interaction implies reciprocity. Thus, NOS, especially nNOS and eNOS, facilitate cavernous engorgement, enabling penile erection and constituting a pivotal aspect in the trajectory of normal male sexual function [42]. Diminished levels of both isoforms may consequently contribute to MI.

In recent years, the new concept of "Male Oxidative Stress Infertility" has emerged [43]. Excessive oxidative stress is a significant factor that can adversely affect sperm health. The imbalance between reactive oxygen species (ROS) and antioxidants is a crucial contributor to MI [44]. When an excessive accumulation of ROS occurs in the body, it can lead to oxidative damage to sperm, potentially resulting in conditions like oligozoospermia or even azoospermia [45]. Key antioxidants, including SOD and GSH-Px, play essential roles in safeguarding cells against oxidative stress [46]. MDA is a product of lipid peroxidation triggered by ROS, and its content serves as an indicator of the extent of oxidative damage to cell membranes [47]. Research findings indicate a notable reduction in SOD and GSH-Px activities within the testes of the model, along with a significant increase in MDA content (refer to Fig 2D). This suggests that the combination of CRS and adenine can lead to heightened oxidative stress in the testes, resulting in oxidative damage that subsequently impacts sperm health [48]. This observation is further supported by the sperm parameters obtained from the model (see Fig 1C and 1D). Moreover, previous studies have demonstrated that CRS can induce severe oxidative stress, leading to diminished antioxidase activity and increased ROS levels [49]. Elevated MDA contents have been detected in the sperm of individuals with conditions such as oligozoospermia or asthenozoospermia [50], indicating a strong association between sperm damage and excessive oxidative stress [8]. Notably, oxidative stress has been closely linked to depression as well [51]. The stress-induced release of proinflammatory cytokines can contribute to various behavioral changes characteristic of depression, including sadness, anhedonia, fatigue, psychomotor retardation, and social withdrawal [52]. Similarly, oxidative stress has been implicated in Kidney Deficiency. For instance, in rats with Kidney Deficiency, SOD activity in serum shows a significant reduction, while MDA content experiences a marked elevation [53].

Depression is often associated with a decrease in monoamine neurotransmitters, such as 5-HT and NE [54]. Therefore, both 5-HT and NE can serve as diagnostic markers for Liver Depression [55]. Following treatment with CRS and adenine, a significant decrease in both 5-HT and NE is observed ($p < 0.01$), as illustrated in Fig 2B. This indicates the manifestation of depressive symptoms in the model group of rats. Moreover, Liver Depression, as characterized by disturbances in emotional regulation, can influence the immune system through neuroendocrine pathways, thereby triggering the production of proinflammatory cytokines, including IL-6, IL-1β, and TNF-α [24], resulting in an inflammatory response. Essentially, the generation of proinflammatory cytokines, acting as immune modulators, is influenced by emotional states, thus participating in the development of Liver Depression [56]. Liver Depression can lead to a decrease in the levels of 5-HT, NE, and an increase in the levels of inflammatory cytokines such as TNF-α and IL-6 [57–59]. Renal depletion of vital essence, colloquially known as

Kidney Deficiency, can engender a decline in intracellular energy reserves and defense mechanisms. This reduction weakens the organism's ability to respond to inflammatory stimuli, making the immune system more susceptible to the influence of proinflammatory cytokines. In this context, intracellular proinflammatory cytokines are prone to generating excessive inflammatory responses, exacerbating the prevailing inflammatory state. As integral reproductive organs, the testes are subject to regulatory control by the immune system and concurrently exhibit interactive influences with LD & KD, which are commonly accompanied by elevated levels of proinflammatory cytokines [53, 60]. Among these cytokines, IL-6, IL-1β, and TNF-α play a pivotal role in provoking testicular impairment [53, 61]. Notably, IL-1β assumes a central role within the cascade of inflammatory responses and can be considered a biomarker for inflammatory reproductive disorders [62, 63], while TNF-α originating from germ cells facilitates testicular detriment through the inflammatory pathways, concurrently dictating the modulation of germ cell apoptosis [64]. Research findings demonstrate that positive expression of proinflammatory cytokines such as IL-6, IL-8, and TNF-α substantially increased within the liver and kidney of the model, as illustrated in Fig 2E. This highlights that CRS and adenine yield a deleterious influence on the organism's immune system, thereby leading to an insufficient regulatory capacity of the liver and kidney towards inflammatory responses, consequently afflicting the testes with damage and perturbing germinal function. This, in turn, has a cascading impact on sperm health, which resonates with the findings graphically presented in Fig 1C, 1D, 1E and 1F.

Liver Depression is commonly associated with disturbances in neural function, and it can disrupt the hormone balance through neuroendocrine pathways such as the hypothalamic-pituitary-gonadal axis (HPGA), leading to an imbalance in the endocrine system [65]. The kidneys play a pivotal role in synthesizing endocrine hormones, including sex hormones. Conversely, Kidney Deficiency could lead to a diminished secretion of hormones, resulting in inadequate sex hormone levels. Therefore, LD & KD can induce endocrine system imbalance through neuroendocrine pathways, thereby affecting hormone synthesis. Concurrently, this endocrine disruption can impact the function of the testes as reproductive organs, thus exerting adverse effects on sperm health. The significance of HPGA in the regulation of male reproduction is undeniable, and the reproductive hormones generated by HPGA play a vital role in orchestrating the production, development, and maturation of sperm, while also holding important significance in the diagnosis and treatment of MI [9]. PRL, a protein hormone secreted by the pituitary gland, when present at elevated levels, can reduce and inhibit androgen receptor expression and activity. This leads to gonadal dysfunction, resulting in ejaculatory anomalies, reduced sperm count, and even azoospermia, thus compromising reproductive function. Within the male body, hypothalamic secretion of GnRH stimulates the anterior pituitary to release gonadotropins, including FSH and LH [66]. FSH plays a crucial role in regulating the division, development, and maturation of spermatogonia within the testes. LH stimulates Leydig cells in the testes to secrete T, which can, under FSH's influence, be converted into $E_2$. T and $E_2$ are indispensable steroid hormones for maintaining normal male reproductive function, and their levels are closely related to spermatogenesis. Consequently, GnRH, FSH, LH, T, and $E_2$ collectively construct a complex regulatory network within the endocrine system, intimately intertwined with testicular reproductive function, collaboratively upholding male reproductive health [67]. Nevertheless, in cases of HPGA dysfunction, disruptions in reproductive hormones may emerge, triggering testicular insufficiency and hindering sperm production, development, and maturation, ultimately culminating in MI. Research findings demonstrate noteworthy reductions in GnRH, T, and $E_2$ levels, coupled with significant elevations in PRL, FSH, and LH levels in the serum of model rats, as detailed in Fig 2A. This suggests that under the combined influence of CRS and adenine, HPGA system

abnormalities and resultant reproductive hormone disarray impair testicular function and diminish sperm quality, the phenomenon further supported by the data presented in Fig 1C, 1D, 1E and 1F.

In summary, the pathogenesis of MI involves a complex interplay of multifaceted factors. LD & KD have emerged as pivotal components intricately linked to the progression of MI. The underlying mechanistic cascade suggests that under conditions characterized by LD & KD, a dysregulation of the body's immune response occurs, triggering the release of proinflammatory cytokines and culminating in an overwhelming state of oxidative stress. Notably, this oxidative stress, coupled with the simultaneous inflammatory milieu, not only attenuates the activity of essential neurological enzymes, thereby impacting penile erection but also exerts deleterious effects on the intricate process of spermatogenesis within the testicular microenvironment. Consequently, a conspicuous decline in sperm quality ensues. Furthermore, the repercussions of LD & KD extend to the endocrine domain, wherein the HPGA encounters duress, leading to disruptions in the delicate balance of reproductive hormones. This, in turn, precipitates a cascade of events that impair testicular function and sperm quality, ultimately culminating in the manifestation of MI [68].

The main focus of this study is to establish a rat model of MI with LD & KD, primarily resulting from the adverse effects of unhealthy liver and kidney function on reproductive capabilities. In other words, liver injury [69], renal dysfunction [70], and oxidative stress [71] may all have detrimental effects on reproductive health. Therefore, this study not only contributes to understanding the mechanisms underlying MI but also provides valuable insights into the impact of liver injury, renal dysfunction, and oxidative stress on reproductive health.

**4.2.2 Urinary metabolic mechanisms of MI and identification of highly associated biomarkers.** Metabolomics, utilizing urine as the investigative specimen, represents a common approach for elucidating the etiology of MI [72]. Research findings illustrate that OPLS-DA effectively distinguishes between the model and the control samples, as depicted in Fig 3D, thereby distinctly revealing significant alterations in the rat's overall metabolic profile following model establishment. Subsequently, KEGG pathway enrichment analysis identified 5 pivotal metabolic pathways: pyrimidine metabolism, alanine, aspartate, and glutamate metabolism, histidine metabolism, D-glutamine, and D-glutamate metabolism, and taurine and hypotaurine metabolism (refer to Table 1 and Fig 3E). Moreover, 23 key biomarkers relevant to MI with LD & KD were pinpointed, where 10 markers exhibited a downregulated tendency, including glutamic acid, alpha-ketoglutarate, L-asparagine, citrate, taurine, cytidine, 4-imidazoleacrylic acid, deoxycytidine, 3-ureidopropionic acid, and 1-methyl-L-histamine, while the remaining 13 markers were upregulated. This is consistent with the metabolic research findings on teratozoospermia [73].

In the plasma of MI patients with Kidney Deficiency, metabolic pathways such as alanine, aspartate, and glutamate metabolism, as well as D-glutamine and D-glutamate metabolism, exhibit manifest anomalies [74], according to the data presented in Table 1. Among the 23 critical metabolites, amino acids constituted the majority, accounting for approximately 47.83% of the total. Additionally, within the 5 pivotal metabolic pathways, 3 were collectively categorized as amino acid metabolism (as shown in Table 1 and Fig 3E). This underscores the intricate association between aberrant amino acid metabolism and the development of MI, implying that disruptions in the reproductive system or sperm impairment are often accompanied by anomalies in amino acid metabolism [75, 76]. This conclusion is supported by other pertinent research; for instance, abnormal amino acid metabolic pathways are evident in the semen of patients with teratozoospermia [77], and levels of glutamine and histidine are notably elevated in the semen of patients with asthenozoospermia in contrast to healthy subjects [78]. Intriguingly, the data outcomes from Table 1 reveal that under the combined influence of CRS

and adenine, glutamine and histidine in urine were upward, whereas glutamate was downward. As the most abundant intracellular amino acid, glutamine is considered to stimulate sperm vitality [79]. From this, it is deduced that the decrease in glutamate levels also acts as an etiological factor contributing to the decline in sperm quality.

In addition to amino acid metabolism, Kidney Deficiency exhibits a notable correlation with aberrant pyrimidine metabolism [80]. Simultaneously, disturbances in pyrimidine metabolism are also evident in the plasma of patients with Liver Depression [81]. This suggests that both Liver Depression and Kidney Deficiency are characterized by anomalies in pyrimidine metabolism, thereby emphasizing the role of pyrimidine metabolism in the pathogenesis of such conditions. Despite taurine and hypotaurine sharing chemical structural similarities with amino acids, the metabolic pathway is commonly categorized under sulfur metabolism. It is widely acknowledged that, on one hand, taurine is an antioxidant, mitigating oxidative damage caused by oxidative stress, thereby contributing to the preservation of sperm integrity and functionality [82]. On the other hand, taurine has the capacity to modulate the synthesis of reproductive hormones through stimulation of HPGA, consequently enhancing sperm quality and vitality [83]. In essence, the decrease in taurine levels can exert detrimental effects on male reproductive function, a proposition substantiated by evidence from Table 1, Fig 1C, 1D, 1E and 1F.

The intersection of animal models and clinical patients reveals 51 shared targets related to MI. Among them, 5 core targets were selected based on the degree: GAPDH, SRC, EGFR, CASP3, and PTGS2, and specific information can be found in Fig 4A and 4B. To explore the interactions between the 23 key biomarkers and the 5 core targets, a molecular docking analysis was performed. The results showed in Fig 4C demonstrate stable binding between the 5 markers (dCMP, dTMP, dUMP, Cytidine, and Deoxycytidine) and their respective targets, characterized by binding free energies of $\leq$ -6.10 kJ·mol$^{-1}$ and a minimum of 2 interaction bonds. This suggests a significant role of the 5 biomarkers in the progression of MI. Specifically, the upregulation of dCMP, dTMP, and dUMP is associated with abnormal energy metabolism, while the downregulation of Cytidine and Deoxycytidine exacerbates oxidative stress, potentially affecting sperm health. Analysis of semen from infertile patients reveals a significant decrease in cytidine levels [84], consistent with the data in Table 1. Notably, all these 5 biomarkers are situated within the pyrimidine metabolism pathway. Earlier discussions have already explored the link between pyrimidine metabolism and LD & KD. Pyrimidine metabolism plays a crucial role in nucleic acid synthesis and protein formation [85]. Under the combined influence of CRS and adenine, pyrimidine metabolism in urine displayed a disrupted state. This further emphasizes that abnormal pyrimidine metabolism stands as one of the primary etiological factors contributing to MI.

The molecular docking results presented in Fig 4C underscore the notable significance of EGFR and PTGS2 as pivotal targets, exhibiting favorable interactions with the 5 biomarkers. The stability of docked complexes and the binding pose obtained in molecular docking is widely validated through MDS [86], as demonstrated by experimental results that show a relatively stable binding between proteins and small molecules. EGFR, a receptor tyrosine kinase, can be activated by various peptide ligands, resulting in the formation of active autophosphorylated receptor homodimers or heterodimers. Serving as a critical regulator of cell growth, survival, proliferation, and differentiation, EGFR plays a vital role in maintaining overall organismal health [87]. Overexpression of EGFR leads to prostate enlargement and impaired reproductive function in male mice [88], along with a positively correlated degree of testicular damage [89]. PTGS2, also known as cyclooxygenase (COX) -2, serves as a key enzyme in prostaglandin synthesis and is closely associated with various diseases like inflammation [90]. Unlike other COX isozymes (e.g., COX-1), PTGS2 is predominantly induced

under stress conditions such as inflammation, rather than under normal physiology circumstances [91]. In cases of testicular inflammation, PTGS2 is significantly upregulated, particularly in the peritubular cells of the testes [92, 93]. Accordingly, under the influence of CRS and adenine, the dysregulated expression of PTGS2 and EGFR in urine disrupts the endogenous metabolite balance, thereby compromising testicular reproductive function and sperm health, ultimately leading to MI. Thus, PTGS2 and EGFR emerge as crucial factors in the pathological mechanisms of MI with LD & KD, potentially serving as promising targets for future therapeutic strategies. Subsequent investigations will further elucidate the potential roles of PTGS2 and EGFR as critical targets within the context of MI with LD & KD, aiming to establish a robust scientific foundation for the targeted development of therapeutic agents and paving the way for precise drug research and development to address this specific form of MI.

## 5. Conclusion

By employing CRS in conjunction with adenine, a clinically representative rat model of MI with LD & KD was successfully established. The model facilitated an exploration of MI pathogenesis from various perspectives, including oxidative stress and endocrine imbalance. Utilizing metabolomics techniques in tandem with bioinformatics analysis, the most significant metabolic pathways, biomarkers, and targets within the urine of MI-induced rats were identified. Furthermore, the comprehensive examination of the overall urinary metabolite levels further provided a profound delineation of the biological mechanisms governing the onset of MI. The characteristics of MI with LD & KD include altered behavior such as increased water consumption, elevated urine output, decreased spontaneous activity, delayed responses, piloerection, aversion to cold, and reduced preference for sucrose water. Additionally, there are significant abnormalities in reproductive function, including decreased sperm count and viability, increased sperm deformity rate, testicular tissue atrophy, and disordered cell arrangement. Hormonal and biochemical imbalances are evident with elevated levels of LH, FSH, PRL, IL-6, IL-1β, TNF-α, and MDA, along with decreased levels of GnRH, T, $E_2$, eNOS, nNOS, SOD, and GSH-Px. Metabolomic analysis reveals changes in specific metabolites and enriched metabolic pathways, while core targets include EGFR and PTGS2. Consequently, this integrated approach furnishes experimental substantiation for the prognostication, diagnosis, and treatment of MI with LD & KD.

## Supporting information

**S1 File. Metabolite names.**
(XLSX)

**S2 File. Screening of targets.**
(XLSX)

**S3 File. Docking information.**
(XLSX)

**S4 File. Multiple indicator measurement data.**
(XLSX)

## Acknowledgments

We extend our sincere appreciation to Guizhou Medical University and Guizhou Yibai Pharmaceutical Co., Ltd. for their invaluable support of this research endeavor.

## Author Contributions

**Conceptualization:** Jian Fan, Ling Tao, Qingbo Yang, Xiangchun Shen.

**Data curation:** Ying Shen, Jian Fan, Ling Tao.

**Funding acquisition:** Qingbo Yang.

**Investigation:** Xiangchun Shen.

**Software:** Ying Shen, Shaobo Liu.

**Supervision:** Xiangchun Shen.

**Writing – original draft:** Ying Shen.

**Writing – review & editing:** Ying Shen, Jian Fan, Ling Tao.

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
