## [Decision Letter · Decision Letter 0]

4 Apr 2024

PONE-D-24-08762Exploring Pathogenesis and Biomarkers Through Establishment of a Rat Model of Male Infertility with Liver Depression and Kidney DeficiencyPLOS ONE

Dear Dr. Shen,

Thank you for submitting your manuscript to PLOS ONE. After careful consideration, we feel that it has merit but does not fully meet PLOS ONE’s publication criteria as it currently stands. Therefore, we invite you to submit a revised version of the manuscript that addresses the points raised during the review process.

Based on the reviewers' suggestions, the paper needs major revision.  The reviewers' comments can be found below.

We look forward to receiving your revised manuscript.

Kind regards,

Tanja Grubić Kezele, Ph.D., M.D.

Academic Editor

PLOS ONE

2. To comply with PLOS ONE submissions requirements, in your Methods section, please provide additional information regarding the experiments involving animals and ensure you have inclTuded details on (1) methods of sacrifice, (2) methods of anesthesia and/or analgesia, and (3) efforts to alleviate suffering.

Reviewers' comments:

Reviewer's Responses to Questions

**Comments to the Author**

1. Is the manuscript technically sound, and do the data support the conclusions?

Reviewer #1: Yes

Reviewer #2: Partly

2. Has the statistical analysis been performed appropriately and rigorously? 

Reviewer #1: Yes

Reviewer #2: No

3. Have the authors made all data underlying the findings in their manuscript fully available?

Reviewer #1: Yes

Reviewer #2: Yes

4. Is the manuscript presented in an intelligible fashion and written in standard English?

Reviewer #1: Yes

Reviewer #2: No

5. Review Comments to the Author

Reviewer #1: The paper `Exploring Pathogenesis and Biomarkers Through Establishment of a Rat Model of Male Infertility with Liver Depression and Kidney Deficiency` discusses the construction of an animal model representing Liver Depression (LD) and Kidney Deficiency (KD) in Male Infertility (MI) through chronic restraint stress (CRS) and adenine treatment. The authors examine ethological and reproductive traits in male rats, along with sperm parameters and testicular/epididymal damage, to mimic clinical features of LD & KD. They delve into the pathogenesis of LD & KD in MI, focusing on oxidative stress, endocrine imbalance, and urinary metabolic mechanisms. Metabolomic analysis identifies potential biomarkers and therapeutic targets for MI.

Strengths:

Multifaceted Approach: The paper employs a comprehensive approach, integrating ethological observations, reproductive assessments, molecular mechanisms, and metabolomics to explore LD & KD in MI.

Relevance to Clinical Practice: By establishing an animal model that mirrors clinical features, the study provides insights into the pathophysiology of MI, potentially guiding diagnostic and therapeutic strategies.

Metabolomic Analysis: The identification of biomarkers and molecular targets using metabolomic techniques adds depth to the study, offering avenues for translational research and personalized medicine.

Weaknesses:

Animal Model Limitations: While the animal model demonstrates parallels to clinical manifestations, it may not fully capture the complexity of human MI. Further validation and refinement of the model are warranted.

Translation to Human Context: The direct translation of findings from animal models to human patients can be challenging due to species differences and other confounding factors. The clinical relevance of identified biomarkers and targets requires validation in human studies.

Metabolomic Data Interpretation: The metabolomic analysis provides valuable insights into metabolic pathways, but the functional significance of identified biomarkers needs to be elucidated. Correlation with clinical outcomes in human populations is essential for validation.

Future Directions:

Validation Studies: Further validation of the animal model and identified biomarkers in human cohorts is crucial to confirm their relevance to human MI.

Clinical Trials: Clinical trials evaluating interventions targeting the identified molecular pathways and biomarkers could provide evidence for personalized therapeutic approaches in MI patients.

Translational Research: Collaboration with clinical researchers and industry partners is essential for translating preclinical findings into clinical applications, particularly in the realm of Predictive, Preventive, and Personalized Medicine (PPPM).

Integration with Existing Studies: The paper should consider integrating findings from related studies (e.g., referenced papers) dealing with similar mechanisms to provide a more comprehensive understanding of MI pathogenesis and potential translational avenues.

Consideration of Related Studies:

Consider series of studies dealing with similar mechanism and suggesting avenues for translation in particular for PPPM

https://link.springer.com/article/10.1007/s13167-017-0115-5

https://link.springer.com/article/10.1186/s13167-015-0034-2

https://link.springer.com/article/10.1007/s13167-017-0098-2

Overall Assessment:

The paper provides valuable insights into the pathogenesis of LD & KD in MI, employing a multifaceted approach encompassing ethological observations, molecular mechanisms, and metabolomics. While the study offers promising avenues for translational research and personalized medicine, further validation in human cohorts and integration with related studies are warranted to enhance its clinical relevance and impact.

Reviewer #2: The authors employed CRS in conjunction with adenine, trying to establish a clinically representative rat model which they named LD & KD in MI. Also, utilizing metabolomics techniques in tandem with bioinformatics analysis, the most significant metabolic pathways, biomarkers, and targets were identified. Overall, this work is a relatively novel study of the male infertility. A few questions and suggestions merit further investigation in this work.

1. I do not know whether the term LD & KD in MI claimed by the author accurately describes the disease state of the animal model. I am confused about whether the disease model is about male infertility, LD, or KD.

2. The author described a large number of contents related to male infertility in the background, but did not clearly describe the relationship between this animal model and male infertility.

3. All pictures do not describe statistical information in detail, such as statistical methods and the amount included in the analysis.

4. The authors claim that they have successfully developed a LD & KD in MI model, but do not specify the accepted criteria for the identification of this model.

6. PLOS authors have the option to publish the peer review history of their article (what does this mean?). If published, this will include your full peer review and any attached files.

Reviewer #1: No

Reviewer #2: No

---

## [Author Response · Author response to Decision Letter 0]

9 Apr 2024

Many thanks to the reviewers, your comments are of great significance to our research and make the manuscript more perfect. Especially regarding the content determination, your valuable comments have made a qualitative leap in the manuscript. Thank you so much. Here, we will answer the reviewers' comments one-on-one.

Reviewer 1：

Comment 1

Strengths: Multifaceted Approach: The paper employs a comprehensive approach, integrating ethological observations, reproductive assessments, molecular mechanisms, and metabolomics to explore LD & KD in MI.

Response 1

This is a very meaningful and instructive comment. Thank you.

Comment 2

Relevance to Clinical Practice: By establishing an animal model that mirrors clinical features, the study provides insights into the pathophysiology of MI, potentially guiding diagnostic and therapeutic strategies.

Response 2

This comment is highly insightful and informative. Thanks.

Comment 3

Metabolomic Analysis: The identification of biomarkers and molecular targets using metabolomic techniques adds depth to the study, offering avenues for translational research and personalized medicine.

Response 3

Your discerning insights into the metabolomic analysis are greatly appreciated. The identification of biomarkers and molecular targets using metabolomic techniques enhances the depth of our study, presenting opportunities for translational research and personalized medicine. We sincerely thank you for acknowledging the significance of this contribution to our work. Thank you.

Comment 4

Animal Model Limitations: While the animal model demonstrates parallels to clinical manifestations, it may not fully capture the complexity of human MI. Further validation and refinement of the model are warranted.

Response 4

Thank you very much for your constructive suggestions, and we will consider implementing them in our subsequent other research endeavors.

Comment 5

Translation to Human Context: The direct translation of findings from animal models to human patients can be challenging due to species differences and other confounding factors. The clinical relevance of identified biomarkers and targets requires validation in human studies.

Response 5

Your suggestion regarding the validation of biomarkers and targets in a clinical setting is not only excellent but also highly practical. We will endeavor to implement such actions in subsequent other relevant clinical studies. Thank you for your proposal.

Comment 6

Metabolomic Data Interpretation: The metabolomic analysis provides valuable insights into metabolic pathways, but the functional significance of identified biomarkers needs to be elucidated. Correlation with clinical outcomes in human populations is essential for validation.

Response 6

We appreciate your insightful comments on the interpretation of metabolomic data. Indeed, the metabolomic analysis offers valuable insights into metabolic pathways. We acknowledge the importance of elucidating the functional significance of identified biomarkers, which necessitates correlation with clinical outcomes in human populations for validation. This step is crucial for enhancing the translational relevance and clinical utility of our findings. Our subsequent other related studies are planned to be conducted following these recommendations.

Comment 7

Future Directions:

Validation Studies: Further validation of the animal model and identified biomarkers in human cohorts is crucial to confirm their relevance to human MI.

Clinical Trials: Clinical trials evaluating interventions targeting the identified molecular pathways and biomarkers could provide evidence for personalized therapeutic approaches in MI patients.

Translational Research: Collaboration with clinical researchers and industry partners is essential for translating preclinical findings into clinical applications, particularly in the realm of Predictive, Preventive, and Personalized Medicine (PPPM).

Integration with Existing Studies: The paper should consider integrating findings from related studies (e.g., referenced papers) dealing with similar mechanisms to provide a more comprehensive understanding of MI pathogenesis and potential translational avenues.

Consideration of Related Studies:

Consider series of studies dealing with similar mechanism and suggesting avenues for translation in particular for PPPM

https://link.springer.com/article/10.1007/s13167-017-0115-5

https://link.springer.com/article/10.1186/s13167-015-0034-2

https://link.springer.com/article/10.1007/s13167-017-0098-2

Response 7

We greatly appreciate your insightful suggestions for future directions. Validating our animal model and identified biomarkers in human cohorts is indeed crucial to substantiate their relevance to human male infertility (MI). Moreover, conducting clinical trials to assess interventions targeting the identified molecular pathways and biomarkers holds promise for advancing personalized therapeutic approaches in MI patients. Collaboration with clinical researchers and industry partners is indispensable for effectively translating our preclinical findings into clinical applications, particularly within the domain of Predictive, Preventive, and Personalized Medicine (PPPM). Additionally, integrating our findings with related studies dealing with similar mechanisms will provide a more comprehensive understanding of MI pathogenesis and offer potential avenues for translation. We will certainly consider a series of studies you suggested regarding similar mechanisms, especially those focusing on PPPM (e.g., referenced papers). These studies are highly informative for our research and provide a solid foundation for our subsequent related investigations. They will further enrich our study and contribute to the advancement of MI treatment strategies.

Reviewer 2：

Comment 1

I do not know whether the term LD & KD in MI claimed by the author accurately describes the disease state of the animal model. I am confused about whether the disease model is about male infertility, LD, or KD.

Response 1

We greatly appreciate your thoughtful consideration of the terminology used in our study. Thank you.

In traditional Chinese medicine theory, "disease" refers to an abnormal state or pathological changes in the body, while "syndrome" is the characteristic pattern of symptoms and signs summarized based on clinical manifestations and physical examinations of patients. "Syndrome" reflects the specific manifestations of the "disease" in the body and is one of the important bases for TCM diagnosis and treatment. A single "disease" may have multiple different "syndrome". In our manuscript, "Male Infertility" is the "disease" we studied, and "Liver Depression" and "Kidney Deficiency" are the most common typical "syndrome" we researched. Therefore, the animal model we prepared is a model with "Liver Depression" and "Kidney Deficiency" as the "syndrome" and "Male Infertility" as the "disease".

Under conditions of negative emotional influence such as depression, anxiety, and sadness, "Liver Qi Stagnation", or "Liver Depression", can be induced. After prolonged restraint (CRS), we observed that all rats eventually gave up struggling, adapting to the "restraint" procedure, and exhibited signs of mild depression. This suggests that Chronic Restraint Stress (CRS) can induce a "Liver Depression" state in experimental rats through its chronic, low-intensity, and long-term stressors. The general behavioral manifestations of rats after modeling include mental lethargy, delayed responses, hair loss, and a tendency to give up struggling, confirming the successful establishment of the "Liver Depression" model. Another important point to note is that the sucrose preference test is an effective method for detecting the disappearance of pleasure response, assessing whether rats exhibit decreased interest and mental lethargy similar to patients with depression. We have elaborated on these aspects in detail in the discussion section of our manuscript. Thus, "Liver Depression" accurately describes the "syndrome" of the animal model we prepared.

"Kidney Deficiency" is a concept in traditional Chinese medicine, referring to the weakness of kidney Qi, which leads to an imbalance in the body's physiological functions. Symptoms such as increased urine volume and decreased spontaneous activity are related to renal function decline caused by "Kidney Deficiency", while manifestations such as squinting, periorbital swelling, erection of the hair, and hunching reflect insufficient kidney Qi. In our study, rats exhibited increased water consumption, increased urine volume, decreased spontaneous activity, squinting, periorbital swelling, erection of the hair, hunching, aversion to cold, sparse body hair, and loss of luster after modeling. These manifestations indicate that these rats exhibited symptoms of "Kidney Deficiency", and "Kidney Deficiency" accurately describes the "syndrome" of the animal model we prepared.

Regarding the "disease" of "Male Infertility", we found that after modeling, the sperm count and sperm viability of rats significantly decreased, while the sperm deformity rate significantly increased, indicating that the spermatogenic function of the rats we prepared was damaged. On the other hand, the levels of nNOS and eNOS in the rat penis significantly decreased after modeling, indicating that the erectile function of the rats we prepared was damaged. Additionally, we conducted sexual behavior monitoring and found that the capture and mounting behavior of rats towards estrous female rats significantly decreased after modeling, indicating a decline in reproductive function in the rats we prepared. These research results collectively demonstrate that we have successfully prepared an animal model of "Male Infertility".

In conclusion, we have successfully prepared an animal model of male infertility with Liver Depression and Kidney Deficiency.

Comment 2

The author described a large number of contents related to male infertility in the background, but did not clearly describe the relationship between this animal model and male infertility.

Response 2

We deeply appreciate the reviewer's insightful observation regarding our manuscript. Indeed, we acknowledge the importance of clearly delineating the connection between our animal model and Male Infertility, especially considering the extensive background information provided. In response to this valuable feedback, we will revise the manuscript to provide a more explicit explanation of how our animal model replicates the key characteristics of Male Infertility. This enhancement will strengthen the scientific rigor and relevance of our study. Thank you for highlighting this crucial aspect, which will undoubtedly enrich the clarity and impact of our research.

Comment 3

All pictures do not describe statistical information in detail, such as statistical methods and the amount included in the analysis.

Response 3

Your astute observation regarding the statistical information in our figures is greatly appreciated. We acknowledge the need to provide more detailed descriptions, including statistical methods and sample sizes, to enhance the transparency and rigor of our analysis. We have carefully revised the figures to include detailed statistical methods and information on sample sizes, aligning with your suggestions. These modifications aim to enhance the clarity and rigor of our analysis. Thank you for highlighting this important aspect of our manuscript.

Comment 4

The authors claim that they have successfully developed a LD & KD in MI model, but do not specify the accepted criteria for the identification of this model.

Response 4

We appreciate the thoughtful review and constructive feedback on our research. Your insights are valuable, and we acknowledge the need for a more detailed explanation of the criteria used to identify the model in our study. Your guidance is instrumental in improving the quality of our research, and we sincerely thank you for your input.

We established an animal model of Male Infertility with Liver Depression and Kidney Deficiency.

Firstly, in terms of behavior, rats in the model group exhibited increased water consumption, elevated urine output, decreased spontaneous activity, delayed responses, piloerection, aversion to cold, and significantly reduced preference for sucrose water. Additionally, there was a significant decrease in the frequency of capturing estrous female rats and mounting behavior.

Secondly, regarding sperm parameters and histopathology, rats in the model group showed a significant decrease in sperm count and viability, along with a notable increase in sperm deformity rate. Many sperm displayed abnormalities such as headless, tailless, and hookless morphology. Moreover, the testicular tissues exhibited considerable atrophy of the seminiferous epithelium, disordered cell arrangement, and loose arrangement of seminiferous tubules.

Thirdly, concerning physiological and biochemical indicators, rats in the model group displayed significantly elevated levels of LH, FSH, PRL, IL-6, IL-1β, TNF-α, and MDA, while GnRH, T, E2, eNOS, nNOS, SOD, and GSH-Px levels were significantly decreased.

Lastly, in terms of metabolites and targets, differential metabolites in the model rats included dCMP, dTMP, dUMP, Cytidine, and Deoxycytidine. Key metabolic pathways were mainly enriched in D-glutamine and D-glutamate metabolism, histidine metabolism, taurine and hypotaurine metabolism, alanine, aspartate, and glutamate metabolism, and pyrimidine metabolism. Core targets included EGFR and PTGS2.

These research findings indicate the successful establishment of animal model of Male Infertility with Liver Depression and Kidney Deficiency. "Male infertility" ("disease") primarily manifested in sperm parameters and histopathology, while "Liver Depression" and "Kidney Deficiency" ("syndrome") mainly manifested in behavior. The specific criteria for assessing Male Infertility with Liver Depression and Kidney Deficiency were mainly reflected in reproductive hormones, inflammatory factors, neurotransmitters, oxidative stress responses, etc., with particular emphasis on differential metabolites, i.e., biomarkers, which are crucial for model evaluation. Additionally, the enriched key metabolic pathways and selected core targets serve as important criteria for evaluating the successful establishment of the animal model. Importantly, these biomarkers and targets lay a scientific foundation for the development of therapeutic strategies and interventions for related diseases in the future.

These points have been supplemented and elaborated in our manuscript. We sincerely appreciate the constructive feedback provided by the reviewer.

---

## [Decision Letter · Decision Letter 1]

22 Apr 2024

Exploring Pathogenesis and Biomarkers Through Establishment of a Rat Model of Male Infertility with Liver Depression and Kidney Deficiency

PONE-D-24-08762R1

Dear Dr. Shen,

We’re pleased to inform you that your manuscript has been judged scientifically suitable for publication and will be formally accepted for publication once it meets all outstanding technical requirements.

Kind regards,

Tanja Grubić Kezele, Ph.D., M.D.

Academic Editor

PLOS ONE

Additional Editor Comments (optional):

Reviewers' comments:

Reviewer's Responses to Questions

**Comments to the Author**

1. If the authors have adequately addressed your comments raised in a previous round of review and you feel that this manuscript is now acceptable for publication, you may indicate that here to bypass the “Comments to the Author” section, enter your conflict of interest statement in the “Confidential to Editor” section, and submit your "Accept" recommendation.

Reviewer #1: All comments have been addressed

2. Is the manuscript technically sound, and do the data support the conclusions?

Reviewer #1: Yes

3. Has the statistical analysis been performed appropriately and rigorously? 

Reviewer #1: Yes

4. Have the authors made all data underlying the findings in their manuscript fully available?

Reviewer #1: Yes

5. Is the manuscript presented in an intelligible fashion and written in standard English?

Reviewer #1: Yes

6. Review Comments to the Author

Reviewer #1: The authors have addressed most of the reviewers' comments. The manuscript can be considered suitable for publication.

7. PLOS authors have the option to publish the peer review history of their article (what does this mean?). If published, this will include your full peer review and any attached files.

Reviewer #1: No

---

## [Editor Report · Acceptance letter]

29 Apr 2024

PONE-D-24-08762R1 

PLOS ONE

Dear Dr. Shen, 

I'm pleased to inform you that your manuscript has been deemed suitable for publication in PLOS ONE. Congratulations! Your manuscript is now being handed over to our production team.

Kind regards, 

on behalf of

Prof. dr. Tanja Grubić Kezele 

Academic Editor

PLOS ONE